# Optimal measurement method for anterior instability on stress radiographs in anterior cruciate ligament tear: Considering the effect of static anterior tibial subluxation

**Joo Hyung Han**[1], **Chong-Hyuk Choi**[2,3], **Min Jung**[2,3], **Kwangho Chung**[2,4], **Sungjun Kim**[5], **Taeho Ha**[3], **Sung-Hwan Kim**[2,3]*

1 Yonsei University College of Medicine, Seoul, Republic of Korea, 2 Arthroscopy and Joint Research Institute, Yonsei University College of Medicine, Seoul, Republic of Korea, 3 Department of Orthopedic Surgery, Severance Hospital, Yonsei University College of Medicine, Seoul, Republic of Korea, 4 Department of Orthopedic Surgery, Yongin Severance Hospital, Yonsei University College of Medicine, Yongin, Republic of Korea, 5 Department of Radiology, Gangnam Severance Hospital, Yonsei University College of Medicine, Seoul, Republic of Korea

* orthohwan@gmail.com

**Data Availability Statement:** All relevant data are within the manuscript and its Supporting Information files.

## Abstract

### Introduction

Accurate assessment of anterior cruciate ligament (ACL) function is vital for guiding treatment. Nevertheless, the presence of tibial subluxation in the neutral position of a patient with an ACL injury may potentially introduce a confounding factor. This study aims to investigate whether tibial subluxation in the neutral position affects the diagnosis of anterior instability in patients with ACL injuries, potentially impacting the reliability and diagnostic accuracy of stress radiography.

### Methods

This study included 88 patients: 30 with acute complete ACL tears (acute group), 28 with chronic complete ACL tears (chronic group), and 30 patients who underwent knee arthroscopic surgery other than ACL reconstruction (control group). Side-to-side differences (SSD) in stress radiography were measured using the Telos load status and the SSD of the gap between the Telos load and unload statuses. Diagnostic accuracy of the two methods was assessed using areas under the receiver operating characteristic curves (AUCs).

### Results

The load SSD (5.92 ± 5.28 mm) was higher than the load-unload SSD (4.27 ± 5.99 mm) in the chronic group ($P = 0.017$). The load SSD demonstrated a significantly higher diagnostic value than that of the load-unload SSD in the combined group (AUC = 0.920 vs. 0.830; $P = 0.012$) and chronic group (AUC = 0.913 vs. 0.754; $P = 0.002$). After adjusting the symptoms for radiographic duration from 6 to 3 months in the chronic group, the load SSD exhibited a

**Funding:** The author(s) received no specific funding for this work.

**Competing interests:** The authors have declared that no competing interests exist.

significantly higher diagnostic value (AUC = 0.902) than that of the load-unload SSD (AUC = 0.740; $P$ < 0.001).

## Conclusion

The load SSD provides superior diagnostic accuracy compared to the load-unload SSD in ACL tear cases, where static anterior tibial subluxation may result in false negatives. Although load-unload SSD may have diagnostic value within the first 3 months post-injury, the load SSD method provides a reliable assessment of ACL function for patients beyond this timeframe.

## Introduction

Evaluation of anterior cruciate ligament (ACL) function is crucial in determining the appropriate treatment for ACL injuries [1–5] and plays a significant role in post-treatment management, particularly in determining the patient's ability to return to their pre-injury activity level [6–8]. While physical examination and patient questionnaires are important for assessing ACL function, there is a need for an objective and quantifiable evaluation tool to ensure more accurate and consistent assessments. Stress radiography using instruments such as Telos and KT-2000 arthrometers has traditionally been employed [9–12], with the recent widespread adoption of the GNRB arthrometer [13, 14].

Among these methods, stress radiography using Telos offers advantages in terms of cost-effectiveness and availability as well as providing a universal and objective assessment of ligamentous knee injuries [15–21]. However, it has some drawbacks, including difficulties in identifying reliable anatomical landmarks and potential errors owing to variations in knee positioning on the Telos device, which can affect the reliability and validity of the measurements [22]. In the measurement of anterior instability using stress radiography, a common approach in clinical practice with the Telos device is to measure the displacement of the tibia under load only [17, 23, 24].

The initial identification of static anterior tibial subluxation following an ACL injury was first proposed by Almekinders *et al.* [25] This phenomenon refers to the abnormal static alignment between the femur and tibia observed on radiographs taken with the knee extended in individuals with insufficient ACL function [26, 27]. Subsequent investigations have shown that this subluxation cannot be reduced and that the normal relationship between the tibia and femur is not restored even after ACL reconstruction [28]. It has been recognized as a significant concern as it can lead to non-anatomic positioning of the tibial tunnel during ACL reconstruction and potentially impact future clinical outcomes [29, 30].

As mentioned above, static anterior tibial subluxation following ACL injury emphasizes the abnormal association between the tibia and femur in individuals with ACL insufficiency. Consequently, static-predisposed tibial subluxation is believed to be a potential confounding factor in the assessment of anterior instability using stress radiography. This raises concerns regarding the reliability of stress radiography as a diagnostic tool.

Preceding studies have noted the necessity of understanding the anterior translation of both the affected and contralateral joints. Joint laxity and the absolute value of anterior translation in each individual joint can influence side-to-side difference (SSD) measurements, thereby affecting the diagnostic process [31, 32]. Furthermore, the degree of anterior translation correlates with the prognosis of conservative treatment for partial ACL ruptures, as well

as influencing surgical outcomes through anterior translation in the contralateral knee [33, 34]. Considering these aspects, the need for a measurement method that considers anterior subluxation in both joints for assessing anterior instability becomes evident.

Therefore, the purpose of this study was to investigate whether static-predisposed tibial subluxation should be considered for the accurate diagnosis of anterior laxity in patients with ACL injury. Specifically, we aimed to compare the diagnostic accuracy of the method we attempted (measuring the difference between the load and unload positions) that reflects tibial subluxation, and the standard method (measuring only the displacement under the load position) to determine the optimal measurement approach for stress radiographs. Additionally, we sought to examine whether there were variations in the diagnostic performance of these two methods based on the chronicity of ACL rupture.

## Materials and methods

This retrospective review included 456 patients who underwent arthroscopic knee surgery at a single tertiary center between January 2020 and January 2022. The study protocol was approved by our Institutional Review Board (IRB 2022-1146-001). The need for obtaining informed consent from participants was waived by the IRB due to the nature of the study involving the analysis of anonymized medical records and archived samples.

The medical records and archived samples for this retrospective study were accessed for research purposes on February 1, 2023. This date marks the initiation of data collection and analysis for the current investigation. To uphold the principles of patient confidentiality and privacy, strict measures were implemented to ensure that the authors did not have access to information that could identify individual participants during or after data collection.

The inclusion criteria for this study were as follows: patients aged > 18 years; patients confirmed to have an acute or chronic complete ACL rupture on MRI; patients with greater than grade II instability on the Lachman and pivot shift tests, which indicated the need for ACL reconstruction due to ACL insufficiency; and patients with a complete ACL rupture confirmed intraoperatively during ACL reconstruction using arthroscopy. The exclusion criteria were as follows: previous surgical history of the affected knee; stiff knee or limited range of motion due to pain; requiring revisional ACL reconstruction; radiographic osteoarthritis of Kellgren-Lawrence grade 2 or higher; concomitant injuries to other ligaments in the affected knee; and concomitant meniscal tears involving displaced portions that caused locking symptoms or the door-stopper phenomenon, such as bucket handle or root tears.

Fifty-eight patients underwent ACL reconstruction and were categorized into two groups based on the criteria commonly used in previous studies [35]: the acute group, which included patients who underwent radiographic imaging within 6 months of injury, and the chronic group, which included patients who underwent radiographic imaging > 6 months after injury. The remaining 340 patients underwent surgeries other than ACL reconstruction.

Ultimately, 30 and 28 patients were enrolled in the acute and chronic treatment groups, respectively. A flow diagram of this study is shown in Fig 1. The control group was adjusted using propensity score matching with variables such as age, sex, BMI, and generalized laxity instead of randomization because this study's retrospective design. There were no significant differences in any of the variables related to patient characteristics (Table 1).

Trained and licensed radiographers performed preoperative stress radiographs of both knees using a Telos® device (Telos GmbH® Laubscher, Holstein, Switzerland) at 150 N. The patient was placed in the lateral decubitus position with the knee flexed at approximately 30˚ (Fig 2A). A pressure plate was positioned at the mid-calf level and counter-bearings were placed at the ankle joint level and approximately 5 cm above the patella. Radiographic

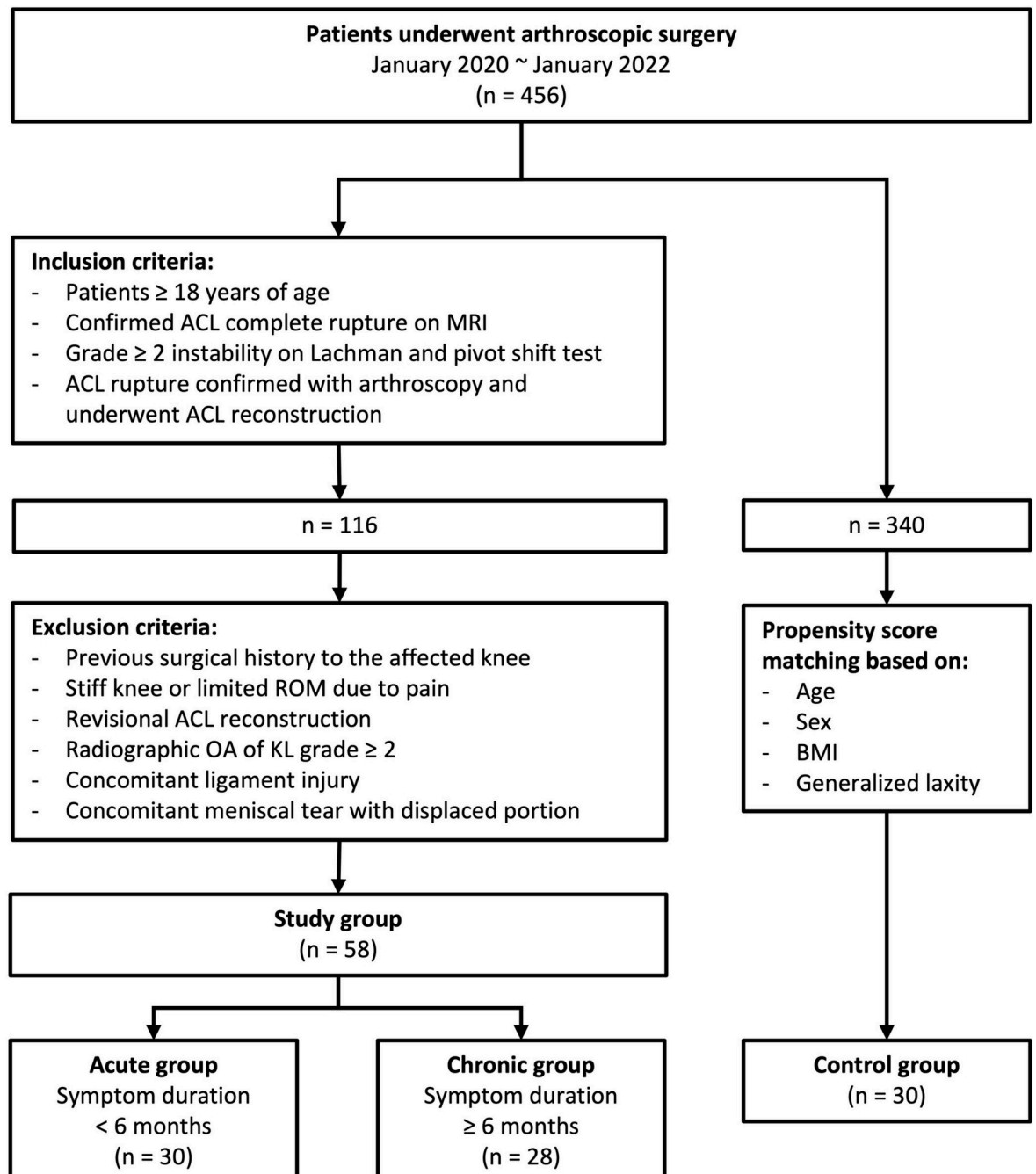

**Fig 1. Flow diagram of this study.** ACL, anterior cruciate ligament; MRI, magnetic resonance imaging; ROM, range of motion; BMI, body mass index.

measurements were performed using Picture Archiving and Communication System (GE Healthcare, Chicago, IL, USA).

A reference line parallel to the medial tibial plateau joint line was used to determine the measurements. Perpendicular lines were drawn tangentially from the reference line to the most posterior contour of the medial femoral condyle and the most posterior contour of the

**Table 1. Summary of the demographic data between the groups.**

| | Acute group (n = 30) | Chronic group (n = 28) | Control group | | | |
| | | | Before matching (n = 340) | *P* value | After matching (n = 28) | *P* value |
| --- | --- | --- | --- | --- | --- | --- |
| Age, y | 32.3 ± 13.8 | 32.4 ± 12.3 | 46.3 ± 17.8 | <0.001* | 32.7 ± 12.2 | 0.807 |
| Sex, n (%) | | | | <0.001* | | 0.467 |
| Male | 27 (90.0%) | 22 (78.5%) | 189 (55.5%) | | 25 (89.2%) | |
| Female | 3 (10.0%) | 6 (21.4%) | 151 (44.5%) | | 3 (10.7%) | |
| BMI, kg/m2 | 24.5 ± 3.2 | 25.4 ± 4 | 26.1 ± 3.6 | 0.032* | 25.0 ± 3.1 | 0.593 |
| General laxity score | 1.5 ± 1.9 | 1.7 ± 2 | 0.4 ± 0.9 | <0.001* | 1.1 ± 1.9 | 0.335 |

*: $P < 0.05$

medial tibial plateau [36, 37]. For the SSD measurements, we employed the center-to-center measurement method using the bone landmarks described above (Fig 2B and 2C). This involved subtracting the value in the unloaded neutral posture from the value in the loaded condition, which is the load-unload method. For the load method, the SSD was calculated while the knee was loaded. These measurements were repeated with a two-week interval by two different orthopedic surgeons. The SSD grading criteria for subgroup analysis were as follows: grade I, < 5 mm translation; grade II, 5–10 mm translation; and grade III, > 10 mm translation [38]. The diagnostic accuracies of both methods were assessed by calculating the area under the receiver operating characteristic (ROC) curve.

Statistical analysis included the following procedures: continuous variables were analyzed using one-way ANOVA, categorical variables were assessed using Fisher's exact test, DeLong's

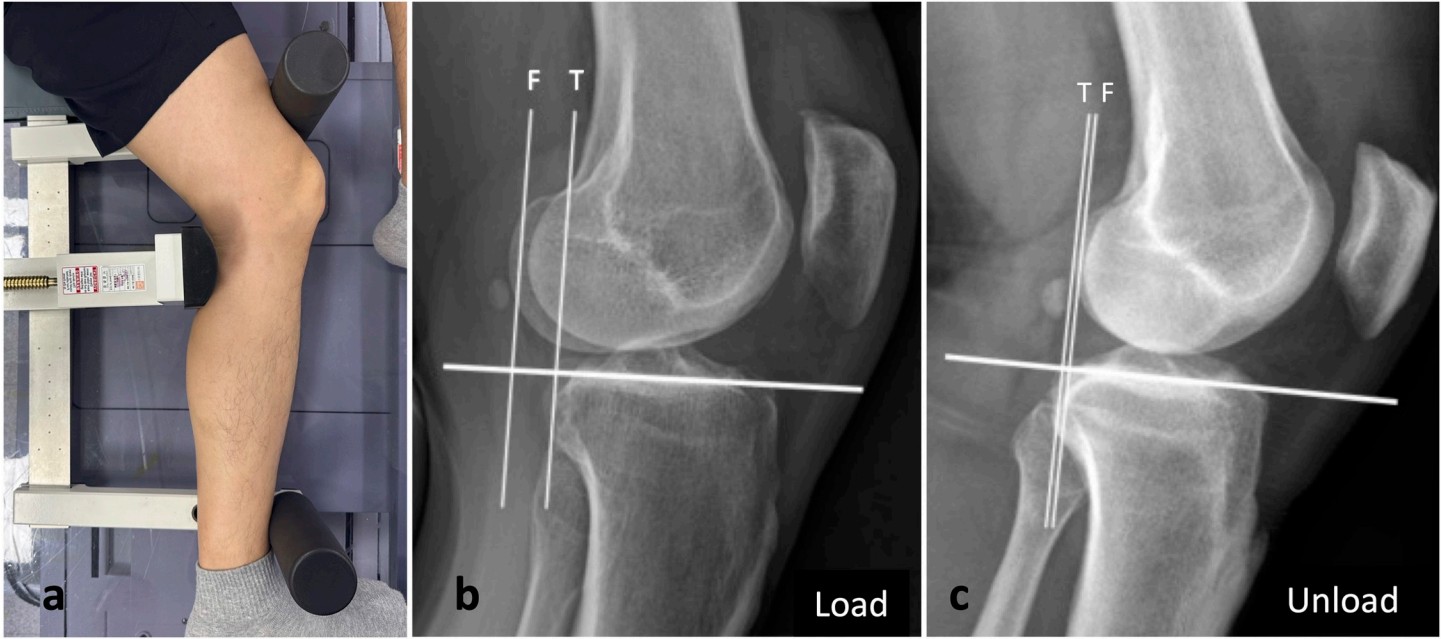

**Fig 2. Patient positioning and anterior instability measurement methods.** (a) shows the patient's position during imaging. The patient was placed in the lateral decubitus position with the knee flexed at approximately 30˚. A pressure plate was positioned at the mid-calf level, and counter-bearings were placed at the ankle joint level and approximately 5 cm above the patella. (b) shows the measurement using the load method, and (c) shows the measurement using the load-unload method. Perpendicular lines were drawn tangentially from the reference line to the most posterior contour of the medial femoral condyle (F) and the most posterior contour of the medial tibial plateau (T). The center-to-center method was used.

test was employed to compare the two ROC curves, and subgroup analysis of SSD was conducted using paired t-tests, Fisher's exact tests, and chi-square tests. Statistical significance was set at $P < .05$. Inter- and intra-observer reliabilities were determined using intraclass correlation coefficients (ICCs).

The sample size for this study was calculated based on previous research with the following conditions: the area under curve (AUC) value of 0.8, a two-tailed α error of 5% and a power (1-β) of 80% was used [39–41]. As a result, a total of 28 patients per group were determined to be the required sample size. All statistical analyses were performed using the R software (version 4.2.1; R Foundation, Vienna, Austria), and data visualization was performed using the ggplot2 package (v3.4.2; Wickham, 2016).

## Results

In the chronic group, the load SSD (5.92 ± 5.28 mm) was significantly higher than the load-unload SSD (4.27 ± 5.99 mm) ($P = 0.017$) (Table 2 and Fig 3). No significant difference was observed in the load SSD (5.86 ± 4.14 mm, -0.67 ± 3.17 mm) and load-unload SSD (5.97 ± 4.16 mm, -0.54 ± 3.96 mm) of the acute and control groups, respectively. Subgroup analyses within each group yielded similar proportions of instability grades. The intra- and inter-observer reliabilities for the radiologic measurements were good to excellent, with an ICC of 0.826 to 0.937.

ROC curve analysis was performed for the acute, chronic, and newly formed combined group consisting of both groups, representing the entire ACL rupture patient population. Patients in each group were categorized as cases, whereas those in the control group were designated as controls. The objective of this analysis was to compare the diagnostic performance of load SSD and load-unload SSD as methods for diagnosing cases.

The load SSD (AUC = 0.920) demonstrated a significantly higher diagnostic value than the load-unload SSD (AUC = 0.830; $P = 0.012$) in the combined group (Fig 4). In the chronic group, the load SSD (AUC = 0.913) exhibited a significantly higher diagnostic value than the load-unload SSD (AUC = 0.754; $P = 0.002$). However, in the acute group, there was no significant difference between the load SSD (AUC = 0.926) and load-unload SSD (AUC = 0.901) ($P = 0.528$).

ROC curve analysis yielded the optimal cutoff values for each group (Table 3). In the combined group, the optimal cutoffs were 5.370 mm for load SSD (specificity:0.893, sensitivity:0.845) and 4.470 mm for load-unload SSD (specificity:0.750, sensitivity:0.828). For the acute group, the optimal cutoffs were 5.612 mm for load SSD (specificity:0.893, sensitivity:0.867) and 4.616 mm for load-unload SSD (specificity:0.929, sensitivity:0.800). In the chronic group, the optimal cutoffs were 5.290 mm for load SSD (specificity:0.893, sensitivity:0.821) and 2.060 mm for the load-unload SSD (specificity:0.750, sensitivity:0.714). Notably, the cutoff value for load SSD remained relatively consistent between 5 and 6 mm between

**Table 2.  Side-to-side difference (mean ± standard deviation) and proportion of instability grade between groups.**

|  | Acute Group (n = 30) | | | Chronic Group (n = 28) | | | Control Group (n = 28) | | |
|---|---|---|---|---|---|---|---|---|---|
|  | **Load SSD** | **Load-unload** | ***P* value** | **Load SSD** | **Load-unload** | ***P* value** | **Load SSD** | **Load-unload** | ***P* value** |
| SSD (mm) | 5.86 ± 4.14 | 5.97 ± 4.16 | 0.862 | 5.92 ± 5.28 | 4.27 ± 5.99 | 0.017* | -0.67 ± 3.17 | -0.54 ± 3.96 | 0.863 |
| Grade I | 17 (56.7%) | 10 (33.3%) | 0.101 | 17 (60.7%) | 18 (64.3%) |  | 27 (96.4%) | 27 (96.4%) |  |
| Grade II | 8 (26.7%) | 16 (53.3%) |  | 6 (21.4%) | 6 (21.4%) | 0.933 | 1 (3.6%) | 1 (3.6%) | 1 |
| Grade III | 5 (16.7%) | 4 (13.3%) |  | 5 (17.8%) | 4 (14.3%) |  | 0 (0.0%) | 0 (0.0%) |  |

SSD, side-to-side difference

*: $P < 0.05$

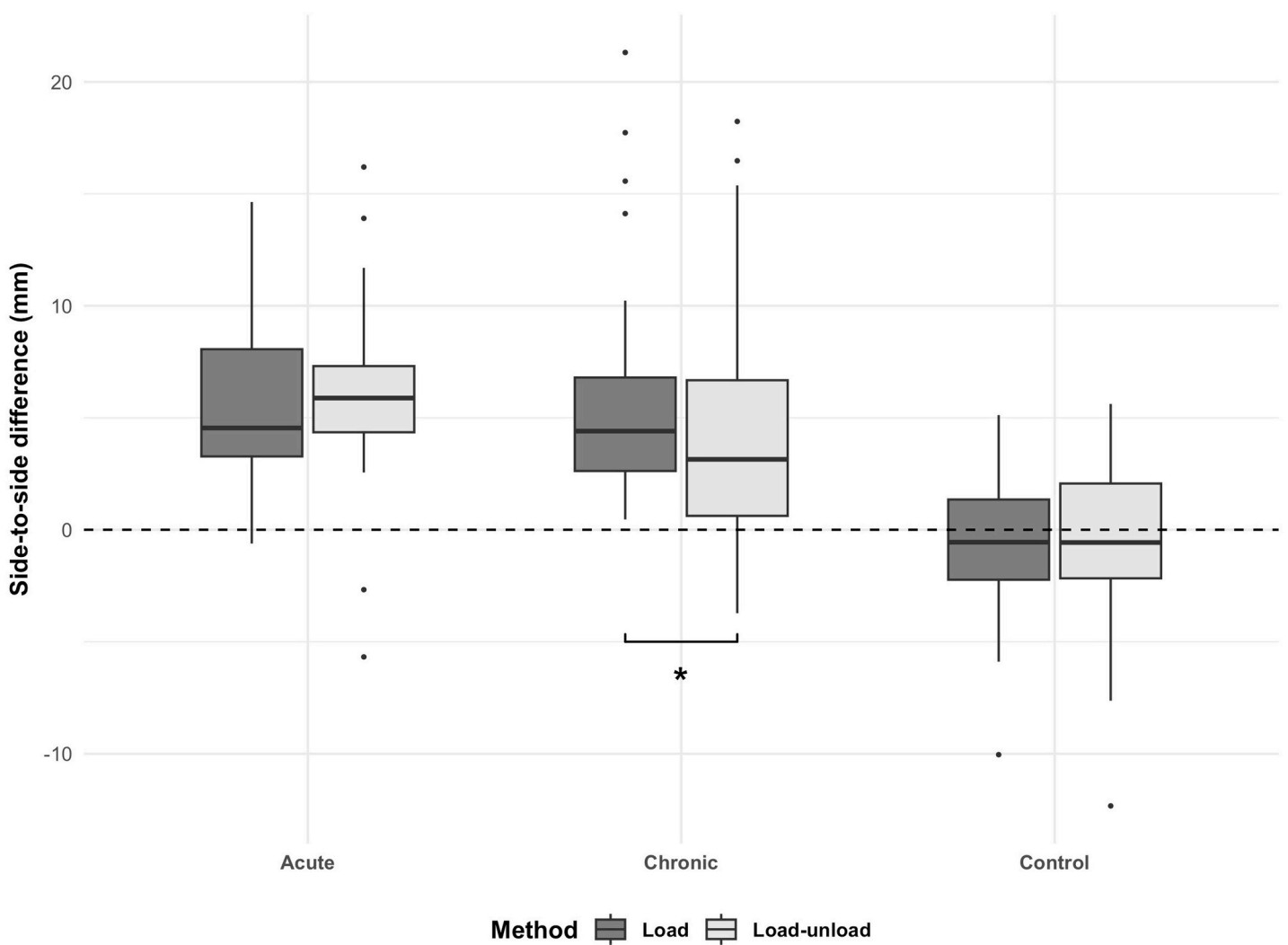

**Fig 3. Box plot of SSD measured using the load and load-unload methods for each group.** In the chronic group, the load SSD was significantly higher than the load-unload SSD. However, no significant difference was observed between the acute and control groups. *: $P < 0.05$.

groups, whereas for load-unload SSD, it was 4.6 mm in the acute group and 2.06 mm in the chronic group.

To assess the diagnostic performance difference, based on the varying cutoff of symptom to radiograph duration, between load and load-unload SSD in the chronic group, we examined the AUC and p-value using the DeLong test for each ROC curve (load and load-unload method) within a range of 7 days to 2 years (Fig 5). Based on the calculated AUC, there was a significant difference in diagnostic accuracy between the two testing methods across the entire range of investigated cutoffs in the chronic group. According to the DeLong test results, the difference in diagnostic accuracy between the two methods was maximized when cutoffs were applied within the range of 90 to 180 days.

Based on the above results, adjusting the reference interval from 6 to 3 months led to changes in the AUC (Fig 6). In the chronic group, load SSD showed a significantly higher diagnostic value (AUC = 0.902) than the load-unload SSD (AUC = 0.740; $P < 0.001$). However, in the acute group, no significant difference was observed between load SSD (AUC = 0.939) and

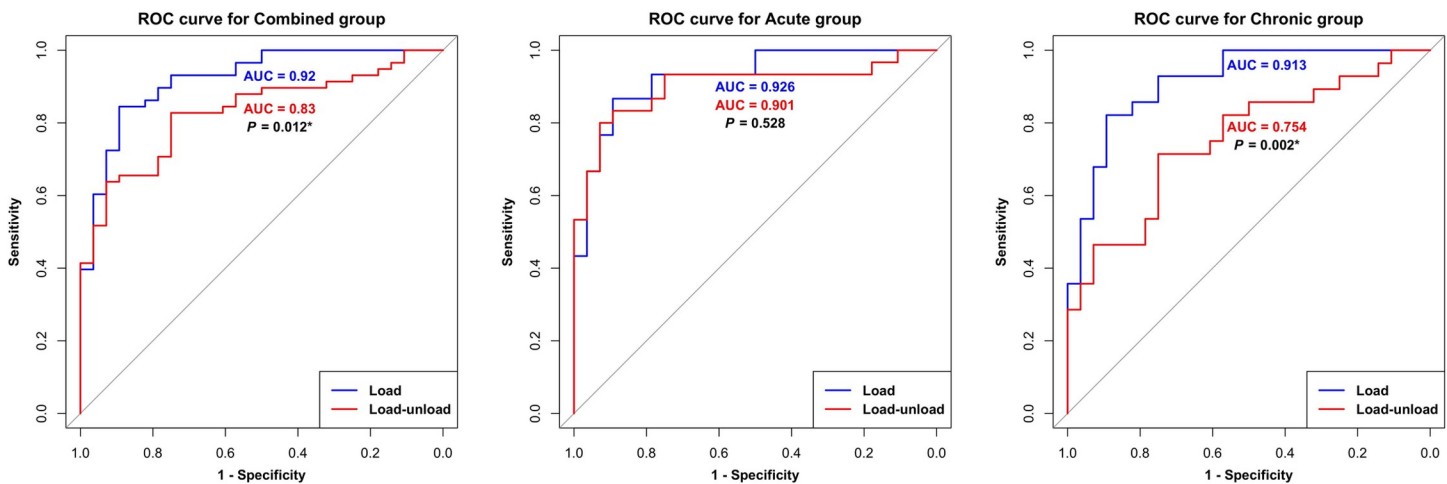

**Fig 4. ROC curve analysis performed for the acute, chronic, and combined groups.** The symptom to radiograph duration, which served as the criterion for distinguishing between the acute and chronic groups, was set at 6 months. In the combined group and chronic group, the load SSD exhibited a significantly higher diagnostic value compared to the load-unload SSD. However, no significant difference was observed in the acute group. ROC, receiver operating characteristic; AUC, area under curve; *: $P < 0.05$.

load-unload SSD (AUC = 0.926) ($P = 0.528$). These findings suggest that when the reference interval was set at 3 months, the load SSD exhibited higher diagnostic power and a more pronounced difference in diagnostic performance between the methods, particularly in the chronic group.

## Discussion

The diagnostic accuracy of the ACL rupture diagnosis, as indicated by the AUC, revealed that the load SSD (AUC = 0.913) demonstrated a significantly higher diagnostic value than the load–unload SSD (AUC = 0.754; $P = 0.002$) in the chronic group. In the case of chronic load-unload SSD, the SSD values sometimes appeared negative, which could be attributed to the influence of static anterior tibial subluxation. Because the measurements were based on the midpoints of the lateral and medial femoral condyles and the medial and lateral tibial plateaus on radiography, it is plausible that anterior tibial subluxation in the lateral compartment also affected the measurement values.

Mixed results have been reported regarding the relationship between fixed tibial subluxation and rupture chronicity in patients with ACL. Almekinders *et al.* [25] were the first to report fixed tibial subluxation, and their subsequent research claimed that osteoarthritic changes in chronically untreated ACL ruptures exacerbate fixed anterior subluxation [42].

**Table 3. Optimal cutoffs and coordinates of the ROC curves.**

| Group | Method | AUC | *P* value | Optimal cutoff, mm | Sensitivity, % | Specificity, % |
|---|---|---|---|---|---|---|
| Combined | Load | 0.920 | 0.012* | 5.37 | 0.845 | 0.893 |
| | Load-unload | 0.830 | | 4.47 | 0.828 | 0.75 |
| Acute | Load | 0.926 | 0.528 | 5.612 | 0.867 | 0.893 |
| | Load-unload | 0.901 | | 4.616 | 0.8 | 0.929 |
| Chronic | Load | 0.913 | 0.002* | 5.29 | 0.821 | 0.893 |
| | Load-unload | 0.754 | | 2.06 | 0.75 | 0.714 |

ROC, receiver operating characteristic; AUC, area under the curve

*: $P < 0.05$

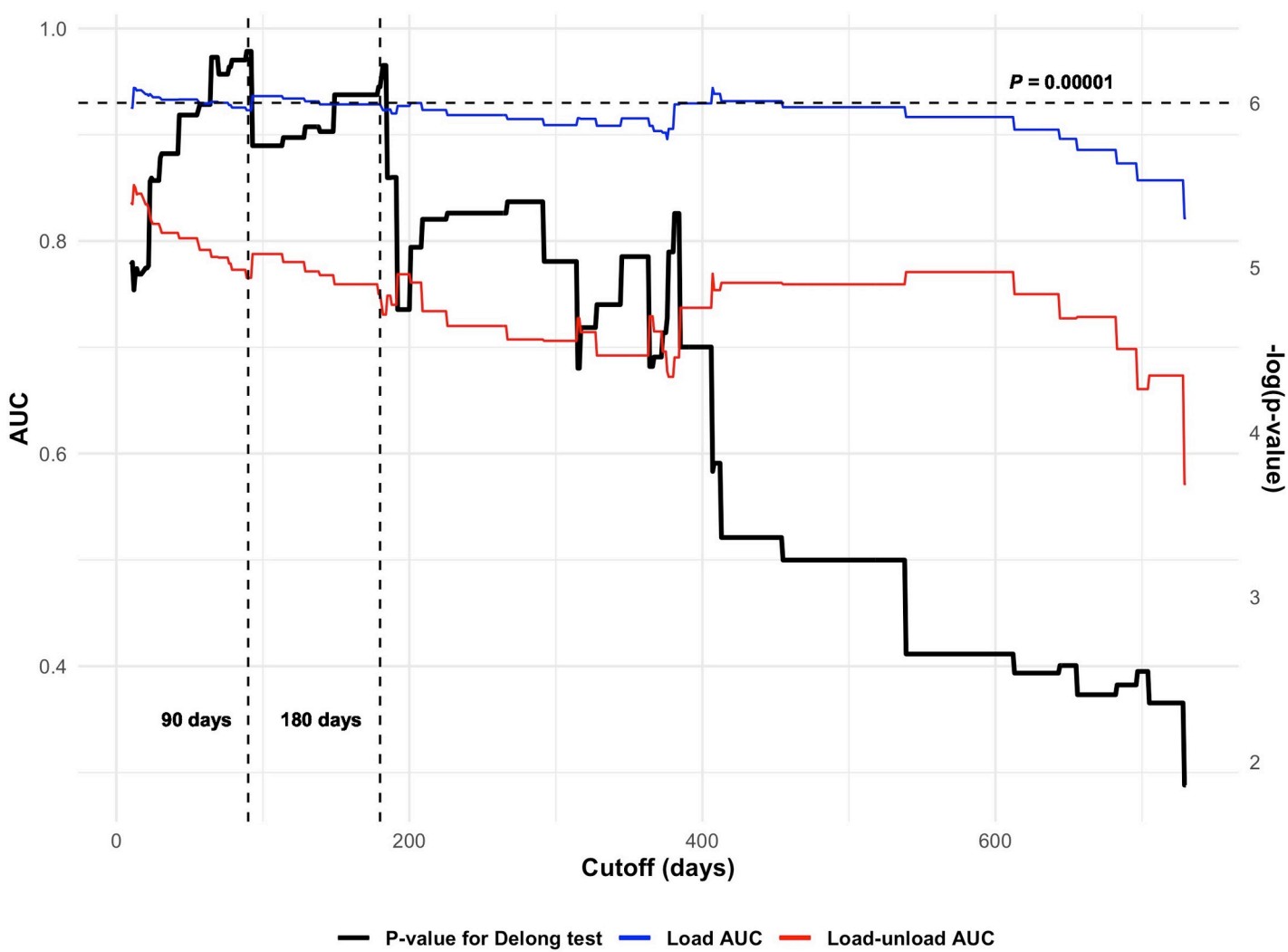

**Fig 5. AUC and p-value of DeLong test comparing each ROC curve (load and load-unload method applied on the chronic group) on varying symptom to radiograph duration applied to define chronic group, with a range of 7 days to 2 years.** The vertical dashed lines represent the respective cutoffs of 90 days and 180 days, and the horizontal dashed line indicates a p-value of 0.00001. The DeLong test results shows the maximized difference in diagnostic accuracy between the two methods within the symptom to radiograph duration range of 90 to 180 days.

However, McDonald *et al.* [29] conducted a comparison of tibial translation between acute and chronic groups to validate these findings, but no significant difference was found between the two groups. It is worth noting that McDonald *et al.*'s study defined the chronic group as patients who underwent knee imaging more than 12 months after an ACL tear, which differs from our study's setting.

High-grade rotatory laxity accompanying ACL rupture results in anterior subluxation of the tibia in both the lateral and medial compartments as well as internal rotation observed on knee imaging [43]. Furthermore, it has been reported that a prolonged time from injury to surgery is a risk factor for high-grade rotatory laxity in chronic patients [44]. The use of radiographs taken in the lateral decubitus position minimizes the influence of gravity, allowing subluxation to persist in this position. Considering the findings of our and previous studies, it can be concluded that this positional adaptation, which is considered a manifestation of rotational instability, may be more pronounced in the chronic group.

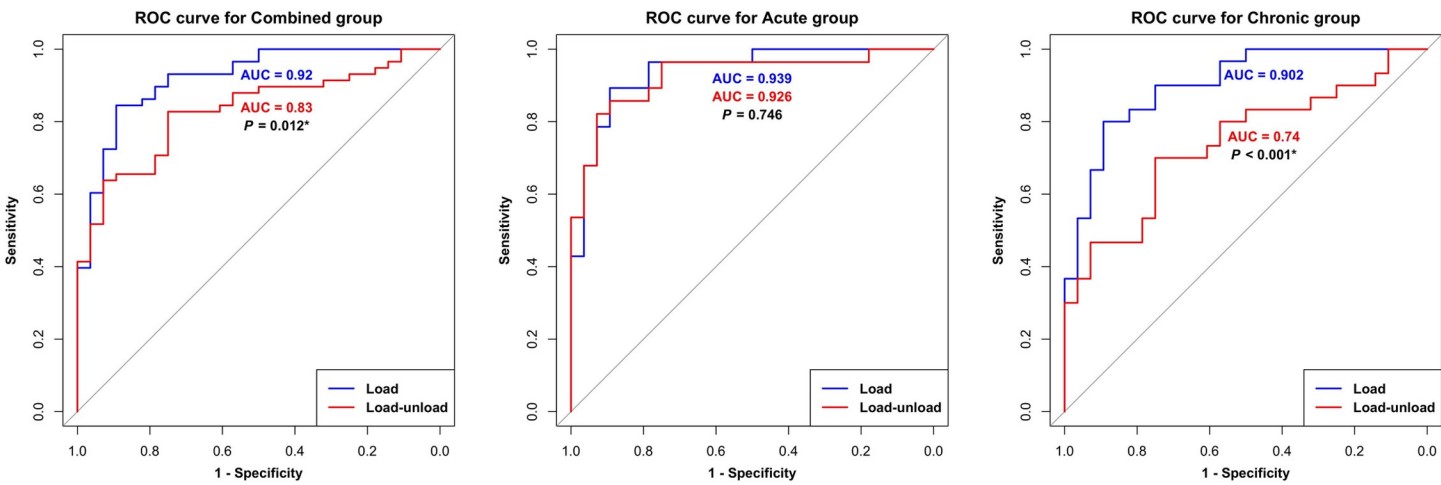

**Fig 6. ROC curve analysis performed for the adjusted symptom to radiograph duration of 3 months based on the Delong test results above.** In the combined and chronic groups, the load SSD exhibited a significantly higher diagnostic value compared to the load-unload SSD and no significant difference was observed in the acute group. The results exhibited a similar trend to when the symptom to radiograph duration was set at 6 months, but a larger difference in diagnostic performance was observed in the chronic group. ROC, receiver operating characteristic; AUC, area under curve; *: $P < 0.05$.

The accuracy of the two methods did not differ in the acute group; however, unloading was clearly inaccurate in the chronic group. The cutoff value remained constant between 5 and 6 mm in load SSD, while in the case of load-unload, it was 4.6 mm in the acute group and 2.06 mm in the chronic group, which is a small value that can also be influenced by measurement errors typically reported to be around 1 mm or lower [17, 24]. Moreover, the cutoff value for SSD used in diagnosing ACL rupture is generally reported to be around 4 to 6 mm [1, 23, 45, 46], which differs from the values observed in the chronic group. Therefore, the relatively low diagnostic ability of load-unload SSD observed in the ROC curve, along with the low cutoff value mentioned above, indicates the difficulty of using load-unload SSD for diagnosis. As reported by Almekinders *et al.*, [42] there is a tendency for anterior tibial subluxation in patients with chronic untreated ACL rupture. Static anterior tibial subluxation itself reduces the difference between the load and unload states of the affected knee, which seems to decrease the diagnostic ability of the load-unload method in the chronic group.

In terms of the chronicity of ACL rupture, as mentioned by DeLee *et al.* [47] there is no specific timeframe for surgical delay, and the physical condition of the patient should be considered rather than a predetermined waiting period when planning the surgical date. Due to this characteristic, the cutoffs used to distinguish between acute and chronic ACL ruptures vary between authors [35]. Some authors use symptom durations such as 6 weeks to 3 months, 6 months, and 12 months or more as cutoffs for chronic ACL ruptures [29, 44, 48–51].

Accordingly, in our study, we initially set the criteria for dividing acute and chronic cases at 6 months. To examine the difference in diagnostic accuracy between the two measurement methods in the chronic group, we conducted AUC and Delong tests by varying the cutoff values. We observed that the difference in diagnostic accuracy between the two methods was maximized when cutoffs were applied within a range of 90 to 180 days. Based on these results, when the reference interval was adjusted to 3 months, load SSD exhibited greater diagnostic power. While there are various criteria for defining chronic ACL rupture from the perspective of diagnosis using stress radiography, this study suggests a symptom duration of 3 months as the cutoff value.

In this study, we compared the method we attempted (measuring the difference between the load and unload positions) that reflects tibial subluxation with the standard method. As mentioned earlier, several other studies have addressed the implications of the anterior

translation of both the affected and contralateral joints [31–34]. However, as demonstrated by the results of this study, the reliability of load-unload SSD is diminished in the chronic status. Therefore, careful consideration is necessary when interpreting the findings, and additional supplementary measurements are warranted to address this limitation.

The limitations of this study were as follows. First, the study had a retrospective design. Second, alternative measurement methods for assessing lateral rotational instability were not used. Further studies are needed to determine whether chronic ACL or anterolateral ligament (ALL) injury is the cause of subluxation. Third, this study was retrospective, and the control group consisted of patients who had undergone different types of knee arthroscopic surgery, rather than healthy subjects. It should be noted that not undergoing ACL reconstruction in the control group does not necessarily imply normal ACL function. Although there may have been patients with poor ACL function in the control group, this is unlikely considering the propensity score-matched average age and other demographics. Lastly, although there may have been issues with randomization in the study design, attempts were made to minimize selection bias through propensity score matching in the control group.

In conclusion, the measurement method for anterior instability using load SSD showed superior diagnostic accuracy compared to the load-unload SSD method, which is susceptible to false negatives owing to static anterior tibial subluxation in ACL tear cases. Within the first 3 months after injury, load-unloading may still provide some diagnostic value. However, for patients beyond this timeframe, an accurate assessment of ACL function can be achieved by comparing both sides using only stress radiographs in the loading state.

## Supporting information

**S1 File. Datasets and R code used in this research.**
(ZIP)

## Author Contributions

**Conceptualization:** Joo Hyung Han, Chong-Hyuk Choi, Min Jung, Sungjun Kim, Sung-Hwan Kim.

**Data curation:** Taeho Ha.

**Formal analysis:** Taeho Ha, Sung-Hwan Kim.

**Investigation:** Joo Hyung Han.

**Methodology:** Joo Hyung Han, Sungjun Kim.

**Project administration:** Sungjun Kim.

**Software:** Joo Hyung Han.

**Supervision:** Sung-Hwan Kim.

**Validation:** Kwangho Chung, Sung-Hwan Kim.

**Visualization:** Joo Hyung Han.

**Writing – original draft:** Joo Hyung Han, Chong-Hyuk Choi, Min Jung, Kwangho Chung, Sungjun Kim, Taeho Ha, Sung-Hwan Kim.

**Writing – review & editing:** Joo Hyung Han, Chong-Hyuk Choi, Min Jung, Kwangho Chung, Sungjun Kim, Taeho Ha, Sung-Hwan Kim.

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
