## [Decision Letter · Decision Letter 0]

21 Jun 2024

PONE-D-23-39613Optimal Measurement Method for Anterior Instability on Stress Radiographs in Anterior Cruciate Ligament Tear: Considering the Effect of Static Anterior Tibial SubluxationPLOS ONE

Dear Dr. Kim,

Thank you for submitting your manuscript to PLOS ONE. After careful consideration, we feel that it has merit but does not fully meet PLOS ONE’s publication criteria as it currently stands. Therefore, we invite you to submit a revised version of the manuscript that addresses the points raised during the review process.

We look forward to receiving your revised manuscript.

Kind regards,

Ismail Tawfeek Abdelaziz Badr

Academic Editor

PLOS ONE

Journal Requirements:

2. In the online submission form, you indicated that information and datasets analyzed during the current study available from the corresponding author on reasonable

request.

Reviewers' comments:

Reviewer's Responses to Questions

**Comments to the Author**

1. Is the manuscript technically sound, and do the data support the conclusions?

Reviewer #1: Yes

Reviewer #2: Yes

2. Has the statistical analysis been performed appropriately and rigorously? 

Reviewer #1: Yes

Reviewer #2: I Don't Know

3. Have the authors made all data underlying the findings in their manuscript fully available?

Reviewer #1: Yes

Reviewer #2: Yes

4. Is the manuscript presented in an intelligible fashion and written in standard English?

Reviewer #1: Yes

Reviewer #2: Yes

5. Review Comments to the Author

Reviewer #1: 1.It is possible that ACL injuries may be combined with other conditions, such as knee osteoarthritis and synovitis. It is therefore important to consider whether the patients enrolled in this study were excluded from the disturbances caused by these possible co-morbidities. Were the patients enrolled in this study with ACL confounders excluded?

2.Whether the two measurements mentioned in the paper, "the traditional load-unload method" and "the load method", are routine in clinical practice.

3.This is a retrospective study, the control group of 28 healthy patients was matched to 340 patients who underwent surgical procedures other than ACL reconstruction, did the surgical operations the patients underwent interfere with the results of this study? These patients also underwent the same assessment components as ACL patients, are these measurements also routine in the clinic?

4.This study concludes,“In conclusion, the measurement method for anterior instability using load SSD showed superior diagnostic accuracy compared to the load-unload SSD method, which is susceptible to false negatives owing to static anterior tibial subluxation in ACL tear cases”,does static anterior tibial subluxation also influence the diagnosis of anterior instability in patients with ACL injuries when using the load-unload SSD method?

Reviewer #2: Firstly, I want to congratulate the authors for the efforts to do this article. Althogut there are limitations, it has a good methodology that support its results. However, I have some considerations and suggestions.

Line 27: "...and 28 healthy patients (control group)". However, in Figure 1, I have observed 30 patients in the control group

Line 92: "...486 patients". However, in Figure 1, I have observed 456 patients

Line 121-126: I suggest you include a figure to better visualize the patient's position for the test. This way, it is easier to reproduce your study.

Conclusion

The purpose of the article in the abstract is "to investigate whether tibial subluxation in the neutral position affects

the diagnosis of anterior instability in patients with ACL injuries, potentially impacting the reliability and diagnostic accuracy of stress radiography". I suggest that you include a clearer answer to this question in your discussion and conclusion.

6. PLOS authors have the option to publish the peer review history of their article (what does this mean?). If published, this will include your full peer review and any attached files.

Reviewer #1: No

Reviewer #2: No

---

## [Author Response · Author response to Decision Letter 0]

8 Jul 2024

We are grateful for the effort that the reviewer has put into the review process. Thanks to the detailed comments provided by the reviewer, we were able to advocate our argument for this paper with improved logical progression. Additionally, in terms of format, we were able to revise the entire manuscript to be more concise and clearer, which has allowed us to convey our conclusion more effectively. Our responses to the reviewer are provided below. Thank you again for your valuable feedback.

GENERAL

Reviewer #1:

1. It is possible that ACL injuries may be combined with other conditions, such as knee osteoarthritis and synovitis. It is therefore important to consider whether the patients enrolled in this study were excluded from the disturbances caused by these possible co-morbidities. Were the patients enrolled in this study with ACL confounders excluded?

-> In this study, factors that could act as confounding variables in measuring anterior instability due to ACL rupture—such as previous surgical history of the affected knee, stiff knee or limited range of motion due to pain, concomitant injuries to other ligaments in the affected knee, and concomitant meniscal tears involving displaced portions—were set as the exclusion criteria.

(Revised page #5/lines 106-111)

As suggested by the reviewer, we also considered osteoarthritis as a factor influencing the patient's symptoms and added it to the exclusion criteria, subsequently reviewing the patients again. The reviewer's advice was very helpful in setting the factors that could influence the interpretation of the results, and we are grateful for that.

(Revised page #5/lines 108-109)

2. Whether the two measurements mentioned in the paper, "the traditional load-unload method" and "the load method", are routine in clinical practice.

-> In the measurement of anterior instability using stress radiography, a common approach in clinical practice with the Telos device is to measure the displacement of the tibia under load only, which we have referred to as the “load method" in this study. This method is routine in clinical practice.

However, static-predisposed tibial subluxation is believed to be a potential confounding factor in the assessment of anterior instability using stress radiography. Therefore, in contrast, the “load-unload method" is the method we attempted, which measures the difference between the load and unload positions (page #4/lines 87-90). This method was used to take into account the degree of tibial subluxation. We have revised the introduction to provide a clearer explanation of these two methods.

(Revised page #3/lines 58-61)

(Revised page #6/lines 135)

3.This is a retrospective study, the control group of 28 healthy patients was matched to 340 patients who underwent surgical procedures other than ACL reconstruction, did the surgical operations the patients underwent interfere with the results of this study? These patients also underwent the same assessment components as ACL patients, are these measurements also routine in the clinic?

-> As you pointed out, this study was retrospective, and the control group consisted of patients who had undergone different types of knee arthroscopic surgery rather than healthy subjects. The measurements used in this study were conducted before surgery, so the operations themselves did not interfere with the results. However, the preoperative diagnoses could have influenced the outcomes. Therefore, it should be noted that not undergoing ACL reconstruction in the control group does not necessarily imply normal ACL function.

To minimize differences in ACL function influenced by factors such as average age and other demographics, we performed propensity score matching. Consequently, considering the average age, we believe the concern is minimal. We have added these points to the limitations.

The measurements used in this study are routinely conducted with the same protocol for patients undergoing knee arthroscopic surgery at the institution where this research was carried out. Therefore, we believe there is little concern regarding the differences in measurement methods among patients included in this study, including the control group. 

These points are important limitations of the study and should be addressed in future research with new designs. We appreciate the reviewer's comments, which allowed us to add this interpretation to the manuscript. Thank you for your valuable feedback.

(Revised page #10/lines 253-266)

4.This study concludes, “In conclusion, the measurement method for anterior instability using load SSD showed superior diagnostic accuracy compared to the load-unload SSD method, which is susceptible to false negatives owing to static anterior tibial subluxation in ACL tear cases”, does static anterior tibial subluxation also influence the diagnosis of anterior instability in patients with ACL injuries when using the load-unload SSD method?

-> To explain the conclusion of this study, it is crucial to understand how much static anterior tibial subluxation influences the diagnostic ability of the load-unload SSD method. As reported by Almekinders et al.,[1] there is a tendency for anterior tibial subluxation in patients with chronic untreated ACL rupture.

Static anterior tibial subluxation itself reduces the difference between the load and unload states of the affected knee, which seems to decrease the diagnostic ability of the load-unload method in the chronic group. Consequently, while the influence on diagnostic ability for acute injuries is small, static anterior tibial subluxation appears to significantly reduce the diagnostic ability of the load-unload method for chronic injuries.

This point has been added to the discussion, and thanks to the reviewer's comment, we were able to include an important argument to support the conclusion of this study. We appreciate this input.

(Revised page #9/lines 234-235)

Reviewer #2:

Firstly, I want to congratulate the authors for the efforts to do this article. Although there are limitations, it has a good methodology that support its results. However, I have some considerations and suggestions.

Line 27: "...and 28 healthy patients (control group)". However, in Figure 1, I have observed 30 patients in the control group

-> The control group consisted of 30 patients, and the content has been corrected.

(Revised page #2/lines 26-28)

Line 92: "...486 patients". However, in Figure 1, I have observed 456 patients

-> As you pointed out, 456 patients is correct. The content has been corrected in Figure 1.

(Revised Figure 1)

Line 121-126: I suggest you include a figure to better visualize the patient's position for the test. This way, it is easier to reproduce your study.

-> A photo of the actual imaging process has been added to visualize the patient's position for the test.

(Revised Figure 2)

Conclusion The purpose of the article in the abstract is "to investigate whether tibial subluxation in the neutral position affects the diagnosis of anterior instability in patients with ACL injuries, potentially impacting the reliability and diagnostic accuracy of stress radiography". I suggest that you include a clearer answer to this question in your discussion and conclusion.

-> As you pointed out, there was a lack of detailed information on how tibial subluxation in the neutral position affects the diagnosis of anterior instability. Almekinders et al.[1] reported that patients with chronic untreated ACL rupture tend to have anterior tibial subluxation. This static anterior tibial subluxation reduces the difference between the load and unload states of the affected knee, which appears to decrease the diagnostic accuracy of the load-unload method in the chronic group. We have added the above content to the discussion to provide a clearer answer on the impact of tibial subluxation.

(Revised page #9/lines 225-230)

The main message of this study was the comparison of the two methods for measuring anterior instability, so this content has been left in the conclusion. Your comments have greatly helped us draw a clear conclusion that aligns with the stated purpose of the paper. Additionally, your detailed feedback allowed us to correct some errors in the manuscript. Thank you for your valuable input.

References

1. Almekinders LC, Pandarinath R, Rahusen FT. Knee stability following anterior cruciate ligament rupture and surgery. The contribution of irreducible tibial subluxation. J Bone Joint Surg Am. 2004;86(5):983-7. doi: 10.2106/00004623-200405000-00014. PubMed PMID: 15118041.

---

## [Decision Letter · Decision Letter 1]

13 Aug 2024

PONE-D-23-39613R1Optimal Measurement Method for Anterior Instability on Stress Radiographs in Anterior Cruciate Ligament Tear: Considering the Effect of Static Anterior Tibial SubluxationPLOS ONE

Dear Dr. Kim,

Thank you for submitting your manuscript to PLOS ONE. After careful consideration, we feel that it has merit but does not fully meet PLOS ONE’s publication criteria as it currently stands. Therefore, we invite you to submit a revised version of the manuscript that addresses the points raised during the review process. Please submit your revised manuscript by Sep 27 2024 11:59PM. If you will need more time than this to complete your revisions, please reply to this message or contact the journal office at plosone@plos.org. Please include the following items when submitting your revised manuscript:A rebuttal letter that responds to each point raised by the academic editor and reviewer(s). You should upload this letter as a separate file labeled 'Response to Reviewers'.A marked-up copy of your manuscript that highlights changes made to the original version. You should upload this as a separate file labeled 'Revised Manuscript with Track Changes'.An unmarked version of your revised paper without tracked changes. You should upload this as a separate file labeled 'Manuscript'.If applicable, we recommend that you deposit your laboratory protocols in protocols.io to enhance the reproducibility of your results. Protocols.io assigns your protocol its own identifier (DOI) so that it can be cited independently in the future. For instructions see: https://journals.plos.org/plosone/s/submission-guidelines#loc-laboratory-protocols. Additionally, PLOS ONE offers an option for publishing peer-reviewed Lab Protocol articles, which describe protocols hosted on protocols.io. Read more information on sharing protocols at https://plos.org/protocols?utm_medium=editorial-email&utm_source=authorletters&utm_campaign=protocols.

We look forward to receiving your revised manuscript.

Kind regards,

Ismail Tawfeek Abdelaziz Badr, M.D.

Academic Editor

PLOS ONE

Journal Requirements:

Reviewers' comments:

Reviewer's Responses to Questions

**Comments to the Author**

1. If the authors have adequately addressed your comments raised in a previous round of review and you feel that this manuscript is now acceptable for publication, you may indicate that here to bypass the “Comments to the Author” section, enter your conflict of interest statement in the “Confidential to Editor” section, and submit your "Accept" recommendation.

Reviewer #1: All comments have been addressed

Reviewer #3: (No Response)

Reviewer #4: All comments have been addressed

2. Is the manuscript technically sound, and do the data support the conclusions?

Reviewer #1: Yes

Reviewer #3: Yes

Reviewer #4: Yes

3. Has the statistical analysis been performed appropriately and rigorously? 

Reviewer #1: Yes

Reviewer #3: Yes

Reviewer #4: N/A

4. Have the authors made all data underlying the findings in their manuscript fully available?

Reviewer #1: Yes

Reviewer #3: Yes

Reviewer #4: Yes

5. Is the manuscript presented in an intelligible fashion and written in standard English?

Reviewer #1: Yes

Reviewer #3: Yes

Reviewer #4: Yes

6. Review Comments to the Author

Reviewer #1: (No Response)

Reviewer #3: This is the second version of this manuscript that was resubmitted after considering the comments of previous 2 Reviewers. The article is about comparing 2 radiological methods for diagnosing the anterior instability in patients with ACL injuries using the Telos device: The load side to side difference (SSD) and the load unload (SSD). The authors found that the load SSD method has a higher diagnostic value specially in chronic cases. They aue this for the static anterior translation occurs in patients with chronic ACL Injuries.

The authors considered all comments of the previous reviewers and modified the article accordingly in a positive way; Yet there are still improvement potential in the article as follows :

• Lines 26-28 (Abstract): the authors included 30 patients in acute group, 28 Patients in the chronic group and 30 patients in the control group. The sum should be 88 not 86 as the authors mentioned.

• Line 90: the authors mentioned that they reviewed 486 patients operated in the their center to choose the included patients, but in the flow chart (Figure 1) they mentioned that the reviewed patients were 456 patients. Please unify this number.

• Line 252: the authors mentioned the abbreviation ALL. Please write it formal because it was not mentioned previously in the . I suppose they mean the anterolateral ligament.

Reviewer #4: Thank you very much for allowing me to review your manuscript. I appreciate the effort you have made performing this study and submitting it to plos one. I think this is appropriate for publishing.

7. PLOS authors have the option to publish the peer review history of their article (what does this mean?). If published, this will include your full peer review and any attached files.

Reviewer #1: No

Reviewer #3: **Yes: **Ayman F. AbdelKawi

Reviewer #4: No

---

## [Author Response · Author response to Decision Letter 1]

14 Aug 2024

We are grateful for the effort that the reviewer has put into the review process. Thanks to the detailed comments provided by the reviewer, we were able to correct the remaining typographical errors in this paper and further improve its overall quality. Our responses to the reviewer are provided below. Thank you again for your valuable feedback.

GENERAL

Reviewer #1: (No Response)

Reviewer #3: This is the second version of this manuscript that was resubmitted after considering the comments of previous 2 Reviewers. The article is about comparing 2 radiological methods for diagnosing the anterior instability in patients with ACL injuries using the Telos device: The load side to side difference (SSD) and the load unload (SSD). The authors found that the load SSD method has a higher diagnostic value specially in chronic cases. They aue this for the static anterior translation occurs in patients with chronic ACL Injuries. 

The authors considered all comments of the previous reviewers and modified the article accordingly in a positive way; Yet there are still improvement potential in the article as follows :

Lines 26-28 (Abstract): the authors included 30 patients in acute group, 28 Patients in the chronic group and 30 patients in the control group. The sum should be 88 not 86 as the authors mentioned.

-> We have corrected this number to 88.

(Revised page #2/lines 26)

Line 90: the authors mentioned that they reviewed 486 patients operated in the their center to choose the included patients, but in the flow chart (Figure 1) they mentioned that the reviewed patients were 456 patients. Please unify this number.

-> The correct number is 456 patients. We have revised the text accordingly.

(Revised page #4/lines 91)

Line 252: the authors mentioned the abbreviation ALL. Please write it formal because it was not mentioned previously in the . I suppose they mean the anterolateral ligament.

-> We intended to refer to the anterolateral ligament. We have revised the text to include both the full term and the abbreviation.

(Revised page #10/lines 254)

Reviewer #4: Thank you very much for allowing me to review your manuscript. I appreciate the effort you have made performing this study and submitting it to plos one. I think this is appropriate for publishing.

---

## [Decision Letter · Decision Letter 2]

2 Sep 2024

Optimal Measurement Method for Anterior Instability on Stress Radiographs in Anterior Cruciate Ligament Tear: Considering the Effect of Static Anterior Tibial Subluxation

PONE-D-23-39613R2

Dear Dr. Kim,

We’re pleased to inform you that your manuscript has been judged scientifically suitable for publication and will be formally accepted for publication once it meets all outstanding technical requirements.

Kind regards,

Ismail Tawfeek Abdelaziz Badr, M.D.

Academic Editor

PLOS ONE

---

## [Editor Report · Acceptance letter]

5 Sep 2024

PONE-D-23-39613R2 

PLOS ONE

Dear Dr. Kim, 

I'm pleased to inform you that your manuscript has been deemed suitable for publication in PLOS ONE. Congratulations! Your manuscript is now being handed over to our production team.

Kind regards, 

on behalf of

Dr. Ismail Tawfeek Abdelaziz Badr 

Academic Editor

PLOS ONE